# Congruence and Complementarity of Differential Mobility Spectrometry and NMR Spectroscopy for Plasma Lipidomics

**DOI:** 10.3390/metabo12111030

**Published:** 2022-10-27

**Authors:** Mohan Ghorasaini, Konstantina Ismini Tsezou, Aswin Verhoeven, Yassene Mohammed, Panayiotis Vlachoyiannopoulos, Emmanuel Mikros, Martin Giera

**Affiliations:** 1Center for Proteomics and Metabolomics, Leiden University Medical Center, 2333 ZA Leiden, The Netherlands; 2Department of Pathophysiology, School of Medicine, National and Kapodistrian University of Athens, 157 71 Athens, Greece; 3Pharmagnose S.A., 320 11 Inofyta, Greece; 4Genome BC Proteomics Centre, University of Victoria, Victoria, BC V8Z 5N3, Canada; 5Institute for Autoimmune Systemic and Neurologic Diseases, 104 31 Athens, Greece; 6Division of Pharmaceutical Chemistry, Faculty of Pharmacy, School of Health Sciences, National and Kapodistrian University of Athens, Panepistiomiopolis, Zografou, 157 71 Athens, Greece

**Keywords:** lipid, lipoprotein, nuclear magnetic resonance, differential mobility spectrometry, mass spectrometry, lipidomics

## Abstract

The lipid composition of lipoprotein particles is determinative of their respective formation and function. In turn, the combination and correlation of nuclear magnetic resonance (NMR)-based lipoprotein measurements with mass spectrometry (MS)-based lipidomics is an appealing technological combination for a better understanding of lipid metabolism in health and disease. Here, we developed a combined workflow for subsequent NMR- and MS-based analysis on single sample aliquots of human plasma. We evaluated the quantitative agreement of the two platforms for lipid quantification and benchmarked our combined workflow. We investigated the congruence and complementarity between the platforms in order to facilitate a better understanding of patho-physiological lipoprotein and lipid alterations. We evaluated the correlation and agreement between the platforms. Next, we compared lipid class concentrations between healthy controls and rheumatoid arthritis patient samples to investigate the consensus among the platforms on differentiating the two groups. Finally, we performed correlation analysis between all measured lipoprotein particles and lipid species. We found excellent agreement and correlation (r > 0.8) between the platforms and their respective diagnostic performance. Additionally, we generated correlation maps detailing lipoprotein/lipid interactions and describe disease-relevant correlations.

## 1. Introduction

Lipids are a class of molecules that is surprisingly large and diverse. Nevertheless, most lipids can be categorized by a building block approach combining a specific head group with fatty acid side chains [1]. Fatty acids exist in various lengths, levels of unsaturation, and functional groups [2,3]. As free carboxylic acids, not combined with a specific head group, they form the lipid class of free fatty acids (FFAs). Among the various possible headgroups, cholesterol (CH), glycerol, and phosphocholine form cholesterol esters (CE), triglycerides (TG), and phosphatidylcholines (PC), respectively, which are prominent and important examples of (plasma) lipid classes [4]. Lipids play essential roles in cell structure, energy metabolism, and signaling [5,6]. The ability to quantify individual lipid molecules is important for understanding these processes, which in turn is vital for the diagnosis, disease prevention, and understanding of disease pathologies. Today, hydrophilic interaction liquid chromatography coupled with either triple quadrupole or high resolution mass spectrometry (MS) and flow-injection-based methods coupled to differential mobility spectroscopy (DMS)-based platforms allow for the quantitative analysis of thousands of individual lipid species among several classes in various biological matrices [4,7,8]. Compared with conventional shotgun lipidomic analysis, DMS allows to separate isobaric species based on their mass, charge, and structure. The most recent development to DMS, the Shotgun Lipidomics Assistant (SLA), a python-based application, facilitates an expanded lipidomics analysis with isotope correction [7]. 

Next to their direct physiological involvement, lipids are also being transported throughout the human body. Thanks to their intrinsic hydrophobicity, they form lipoproteins [9], which facilitate their transport in the bloodstream. Lipoproteins are droplets of mostly lipids and, to a lesser extent, proteins that allow the hydrophobic lipids to participate in the circulation. Lipoproteins can be grouped into various classes and subfractions, depending on their function, density, protein content, and size [10]. In recent years, NMR has become available as a relatively straightforward method to analyze lipoproteins in blood plasma/serum in great detail [11]. Just recently, an NMR-based atlas of lipoprotein-related health markers obtained from almost 120,000 individuals has been published [12]. The authors successfully correlate specific lipoprotein parameters to several diseases such as, for example, lung cancer, sepsis, and depression. Still, NMR-based lipid and lipoprotein analysis is restricted to lipid class quantification without granting further details into lipid fine structures, as can be achieved by MS-based approaches. However, in contrast with MS, NMR can analyze plasma/serum non-destructively and leaves assemblies like lipoproteins intact. On their own, both MS-based lipid analysis and NMR-based lipoprotein analysis are of great value to our understanding of lipid biology [13,14,15,16]. In this report, we will evaluate the quantitative agreement of NMR and DMS-SLA analysis for lipid class quantification and investigate how the two methods can complement each other in order to obtain an even broader and better understanding of patho-physiological lipoprotein and lipid alterations. 

To do so, we defined three distinct aims of the presented study: (i) to compare the quantitative performance for lipid class quantification between our internationally benchmarked DMS-SLA-based lipidomics platform [4,7] and the NMR-based Bruker in vitro diagnostics for research (IVDr) lipoprotein subclass analysis (B.I.LISA) service as part of their platform [17], (ii) to develop a combined sample preparation work flow making use of the non-destructive nature of NMR sample analysis, and (iii) to correlate lipoprotein-related parameters with lipid class and fine structure analysis. To address sample variability and inter-individual variance, we investigated complex lipids and lipoproteins from standardized control plasma as well as a cohort of rheumatoid arthritis (RA) patients in comparison with healthy controls. 

## 2. Materials and Methods

### 2.1. Chemicals and Consumables

All chemicals and consumables were of LC-MS grade or higher. Methanol was purchased from Merck, Darmstadt, Germany. Dichloromethane (DCM), methyl-*tert*-butyl ether (MTBE), 1-Propanol, and LC-MS grade water were purchased from Honeywell, Germany. Ammonium acetate was purchased from Sigma-Aldrich, United Kingdom. Lipidyzer internal standard (IS) kits covering 54 deuterated lipid species from 13 major lipid classes were purchased from Sciex, Framingham, MA, USA. Additionally, 16 deuterated lipid species from phosphatidic acid (PA), phosphatidylserine (PS), phosphatidylinositol (PI), and phosphatidylglycerol (PG) lipid classes produced by Avanti Polar lipids were purchased from Merck, Germany. For details about the DMS-SLA method, please refer to Ghorasaini et al. [4] and Su et al. [7]. The sodium 3-[trimethylsilyl] d4-propionate (TSP) internal NMR reference was purchased from Cambridge Isotope Laboratories, Tewksbury MA, USA. The disodium phosphate (Na_2_HPO_4_), sodium hydroxide, and sodium azide for the NMR buffer solution were purchased from Sigma-Aldrich, St. Louis MI, USA.

### 2.2. Collection of Plasma Samples

In this study, 54 RA patients and 11 healthy volunteers from the Greek population were recruited as part of the ArthritisHeal project at the Laikon general hospital in the outpatient clinic, after obtaining written informed consent. Plasma samples from RA patients, fulfilling the ACR/EULAR [18] criteria at diagnosis, and healthy controls were collected pre-prandially in the morning in K2 ethylenediaminetetraacetic acid-containing (K-EDTA) vacutainers, following overnight fasting. Plasma samples were obtained after centrifugation at 1500× *g* for 10 min at 4 °C and were stored at −80 °C within 30 min. Information about the study participants including age, gender, auto-antibody status (serological markers), current therapy, and comorbidities was reported. RA patients included those receiving DMARDs, biological DMARDs, or a combination of both. The collection of human body fluids from healthy and RA patients was approved by the Bioethics and Deontology Committee of the Medical School of the National and Kapodistrian University of Athens (study protocol 296/21-05-2020), as well as the Scientific Council of the General Hospital of Athens, “LAIKO” (study protocol 8330/19-05-2020). All participants signed a written informed consent form prior to enrolment.

### 2.3. Sample Preparation

For lipoproteins’ analysis by NMR spectroscopy, the sample preparation was performed according to the requirements of the Bruker B.I.LISA lipoprotein analysis protocol. Here, 300 µL of EDTA plasma samples was mixed with 300 µL of 75 mM disodium phosphate buffer in H_2_O/D_2_O (80/20) with a pH of 7.4 containing 6.15 mM NaN_3_ and 4.64 mM sodium 3-[trimethylsilyl] d4-propionate (TSP) using a Gilson 215 liquid handler in combination with the Bruker SampleTrack system. Next, the samples were transferred into 5 mm SampleJet NMR tubes in 96-tube racks with a modified second Gilson 215 liquid handling robot. Finally, the tubes were closed by POM ball insertion. While queued for acquisition, the samples were kept at 6 °C on a SampleJet sample changer.

After completion of NMR measurements, 50 µL of sample-buffer mixture from the NMR tubes was aliquoted in 2.0 mL Eppendorf safe-lock tubes for lipid extraction with MTBE, as described by Matyash et al., with some modifications [19]. Then, 25 µL IS mix, 575 µL MTBE, and 150 µL methanol were added to the tubes and vortexed. The mixture was kept at room temperature for 30 min and then centrifuged for five minutes at 18,000× *g*, 20 °C. Next, 750 µL of supernatant was transferred to a new 2.0 Eppendorf safe-lock tubes. Extraction was repeated on the original tubes with 300 µL MTBE and 100 µL methanol. Then, 350 µL of supernatant from the second extraction was transferred to the new Eppendorf. Then, 300 µL LC-MS grade water was added for phase separation and organic supernatants were collected after centrifugation for five minutes at 18,000× *g*, 20 °C. Afterwards, 650 µL of the upper organic layer was transferred to a glass vial and dried under a mild stream of nitrogen. Next, the samples were reconstituted with 250 µL of DMS-SLA running buffer (10 mM ammonium acetate in dichloromethane: methanol (50:50 vol/vol)) and transferred to an glass vial for DMS-SLA analysis.

### 2.4. NMR Experiments and Data Processing

A 600 MHz Bruker Avance Neo spectrometer (Bruker Corporation, Billerica, MA, USA) was used to perform proton nuclear magnetic resonance (^1^H-NMR) experiments. The spectrometer was equipped with a 5 mm triple resonance inverse (TCI) cryogenic probe head complemented by a Z-gradient system and automatic tuning and matching [20]. The NMR spectra were acquired following the Bruker B.I.Methods protocol. The duration of the π/2 pulses was calibrated automatically for all individual samples with a homonuclear-gated nutation experiment on the locked and shimmed samples [21]. Temperature calibration was performed before the experiments using a standard 5 mm sample of 99.8% methanol-d4 (Bruker). All experiments were recorded at 310 K. A standard 5 mm QuantRefC sample (Bruker) was measured as the quantification reference and for quality control. For water suppression during the relaxation delay and the mixing time of the NOESY1D experiment, pre-saturation of the water resonance with an effective field of γB_1_ = 25 Hz was applied [22]. The NOESY1D experiment was recorded using the first increment of a NOESY pulse sequence [23]. Using a relaxation delay of four seconds and a mixing time of 10 milliseconds, 32 scans of 98,304 points covering a sweep width of 17,857 Hz were recorded after applying four dummy scans. The NOESY1D spectra were submitted to the commercial Bruker NMR platform to extract the lipoprotein concentration values. 

### 2.5. DMS-SLA Experiments and Data Processing

DMS experiments were carried out using SLA [7], which is essentially an extended open access version of the Sciex Lipidyzer platform [4]. In short, a SCIEX QTRAP 5500 mass spectrometer equipped with SelexION DMS interface and Nexera X2 UHPLC-system operated with Analyst software was used. The samples were analyzed using multiple reaction monitoring (MRM) in two consecutive flow injection analysis (FIA) methods with positive and negative ionization mode. Here, 75 µL of the reconstituted samples was injected using Shimadzu SIL 30AC autosampler into the running buffer at an isocratic flow rate of 8 µL/min. After six minutes, the flow rate was ramped to 30 µL/min for two minutes for washing. In the first method, PC, lysoPC (LPC), phosphatidylethanolamine (PE), lysoLPE (LPE), Sphingomyelin (SM), PS, PI, and PG were separated with the SelexION DMS cell. To improve DMS separation, 1-propanol was added to the curtain gas as a chemical modifier. In the second method, FFA, TG, diacylglycerides (DG), ceramides (CER), dihydroceramides (DCER), lactosylceramides (LCER), hydroxyceramides (HCER), CE, and PA were measured with the DMS-cell switched off. SLA software was used to process data files and report the lipid class and species concentration and composition values. For further details including extensive lists with all instrumental settings (e.g., mass spectrometric transitions, internal standards, and ionization modes), please refer to Ghorasaini et al. [4] and Su et al. [7]. To allow platform comparison, the concentration unit nmoL/mL reported by the SLA software was converted to mg/dL using the formula (nmoL/mL × molar mass of lipid)/10^4^. 

## 3. Results and Discussion

As outlined above, we had three main goals: (i) comparison of the quantitative performance between NMR and DMS-SLA, (ii) development of a combined work flow, and (iii) correlation of lipoprotein (sub)fraction distribution and lipid fine structures. With respect to (i), three major lipid classes—CE, phospholipids (PL), and TG—overlap between the platforms and we conducted a comparative analysis. In addition, we compared lipid class concentrations between healthy controls and RA patients to investigate the consensus among the platforms on differentiating the two groups. With regards to (ii), as NMR allows for non-destructive sample analysis, we utilized a combined workflow for consecutive NMR and DMS-SLA measurements on single sample aliquots. As pH stabilization is crucial to the performance of NMR, samples are mixed 1:1 with disodium phosphate buffer prior to analysis. The as-prepared samples can subsequently undergo comprehensive lipid analysis using DMS-SLA. In turn, buffer addition is the main characteristic of a combined workflow that had to be investigated. To do so, we analyzed patient and quality control samples with and without buffer addition. Ultimately, (iii) we performed a correlation analysis between all lipoprotein particles and lipid species measured in healthy controls and RA patient samples.

### 3.1. Comparison of NMR and DMS-SLA Platform

We measured a total of 112 lipoprotein particle-related parameters using the NMR platform. More specifically, these parameters consist of TG, PL, total CH, free CH (FC), Apolipoproteins A1/A2/B100, and the B100/A1 ratio in different density classes. CE are not part of the standard set of variables, but these were calculated as the difference between total CH and FC. Similarly, we detected 755 lipid species representing 17 lipid classes with the DMS-SLA platform. The concentration of lipid species was summed up to obtain total lipid class concentrations. Of these many variables provided by the two platforms, we compared the measurement outcomes of CE, TG, and PL lipids.

Method comparison is frequently carried out using correlation and regression analysis. However, these methods do not study the methodological bias and do not adequately assess method agreement. When two methods are designed to measure identical variables, they should have a linear relationship with a good correlation. Bland and Altman (BA) in 1983 re-proposed an approach, firstly presented by Eksborg in 1981 [24], to compare two quantitative measurements by studying the mean difference and constructing the limits of agreement [25]. A BA plot is essentially a XY scatter plot, where the Y-axis and X-axis represent the difference and average of the compared methods, respectively. We utilized both BA and correlation analysis in this study.

From BA analysis, we found excellent agreement between the platforms with a bias of 1.1%, −2.2%, and 8.0% for PL, TG, and CE concentrations, respectively (Figure 1A). The intervals of agreement were in the range of or below 10% of the mean concentration values for all three lipid classes. Furthermore, we found good correlation among the platforms with no obvious deviations from linearity (Figure 1B). Ideally, when two measurements are in absolute agreement, the mean difference (bias) and the standard deviation (SD) of the difference would be zero. A positive bias indicates that the DMS-SLA method reports higher concentration values and a negative bias indicates that the NMR method reports higher concentration values. In our case, we found a small positive bias for CE and PL measurements and a small negative bias for TG measurement. With a BA plot, it is also possible to estimate an agreement interval within which 95% of the differences fall. The BA method recommends that 95% of the measurements should lie within ±2SD of the mean difference to consider method agreement. Here, we found >95% of the measurements within the limits of agreements for all three lipid classes.

Pearson’s (r) and Spearman’s (ρ) correlation analysis showed excellent correlation among the platforms for both original (mg/dL) and log-transformed concentration values (Table 1). The correlation coefficients r and ρ were above 80% with *p* < 0.05 for all three lipid classes. Furthermore, we calculated the root-mean-square deviation (RMSD) and normalized RMSD (NRMSD) values to investigate if the correlation models predict the data accurately. We found good NRMSD values of 14.82%, 7.77%, and 9.44% for TG, CE, and PL correlation models, respectively. Although NMR and DMS-SLA are entirely different analytical platforms with their own specific workflow and technology, we found excellent agreement and correlation among the reported lipid class concentrations. 

### 3.2. Effect of Buffer Addition on DMS-SLA Measurements

NMR analysis requires dilution of plasma samples with phosphate buffer to stabilize the pH and the addition of a reference compound (TSP). Apart from this, NMR analyzes the sample non-destructively and, after analysis, the samples can be used for other purposes. Using the NMR sample for DMS-SLA analysis could be problematic owing to the presence of inorganic salts that potentially decrease the sensitivity of electrospray ionization (ESI) in MS analysis [26] and possibly disturb the lipid extraction process. In order to investigate to which extent the NMR buffer interferes with the DMS-SLA analysis, we analyzed two sets of 10 patients and three quality control plasma samples with and without buffer addition. The lipid species concentration of Cer d18:0, PS, and PG lipids in plasma were below the blanks, hence they are not included in this analysis. We found excellent agreement and correlation between the lipid concentrations measured in these samples (Figure 2) (BA plots can be found in Appendix A). The Pearson’s correlation coefficient (r) and the slope of the regression line prove the agreement among the two sample sets and confirm that buffer addition does not significantly influence DMS-SLA analysis.

### 3.3. Correlation between Lipids and Lipoprotein Particles

The endogenous lipoprotein pathway facilitates transport of TG and CH synthesized in the liver to peripheral tissues (Figure 3A). Very-low-density lipoprotein (VLDL) particles are the major transporter of TG from the liver to peripheral tissues, where they are hydrolyzed into FFA and glycerol by lipoprotein lipase [27]. From correlation analysis, we found a strong positive correlation between VLDL particles and TG lipids (Figure 3B) (full names of lipoprotein abbreviations are available in Appendix A and the interactive heatmap in Appendix A). The correlation was stronger for the TG lipids with a smaller total carbon number, indicating that, when the total plasma TG content increases, the average alkyl chain length decreases. Similarly, a strong negative correlation was observed between VLDL particles and FFA. Considering that FFAs are one of the major substrates for hepatic VLDL-TG production [28] and the product of peripheral VLDL-TG hydrolysis, a negative correlation between them makes sense.

The release of TG from VLDL particles results in the formation of VLDL remnants called intermediate-density lipoprotein (IDL). IDL particles still contain a considerable amount of TG, which is further hydrolyzed. Similar to VLDL particles, we observed a strong positive correlation between IDL particles and TG lipids (Figure 3B). Further hydrolysis of TG from IDL particles decreases the TG content and CE-enriched low-density lipoprotein (LDL) particles are formed. Interestingly, we observed a positive correlation between the TG lipids and the TG content of the larger LDL particles and the number of large LDL particles, but the correlation is much lower between the TG lipids and the TG content of the smaller LDL particles and other LDL components. The number of small LDL particles even decreases with a larger total plasma TG content. This means that a higher plasma TG content results in a greater number of large LDL particles with a larger TG content relative to the other LDL components. NMR does not measure CE lipids directly; therefore, the correlation between CE lipids and LDL particles cannot be addressed. However, a strong positive correlation was observed between the LDL-CH lipid class and CE lipids, consistent with the higher cholesterol content of LDL particles relative to VLDL.

High-density lipoprotein (HDL) is enriched in CE and PL and plays an important role in reverse CH transport from peripheral tissues to the liver [27]. The nascent HDL synthesized by the liver picks up free CH from the cells and FFAs from lecithin by lecithin-cholesterol acyl transferase to form a mature CE-enriched HDL. We observed a strong positive correlation of HDL-CH and HDL-PL sub-particles with CE and PL lipids, respectively. On the other hand, we observed a strong negative correlation of TG lipids with HDL particle components, except for the HDL TG content. Of the subfractions, the TG content of the smaller HDL particles shows the strongest correlation. A higher total plasma TG content results in a higher HDL TG content, with this TG mainly being transported in smaller HDL particles. Intriguingly, a higher level of TG unsaturation correlates with more TG in larger particles. To summarize, the correlation between lipids and lipoproteins was dependent on the lipid composition in the center core of lipoproteins, and these results support the congruence among the NMR and DMS-SLA platforms. In addition, comprehensive investigation of the carbon chain length and saturation of lipids with lipoprotein sub-particles could provide better insights into the lipid–lipoprotein interaction.

### 3.4. Comparative Analysis among Healthy Control and RA Patient Plasma

To assess the useability of both platforms in distinguishing biomaterials from healthy and diseased subjects, we compared performance using an RA plasma cohort in a patient control setting. We assessed total TG, CE, and PL plasma levels in 11 healthy controls and 21 RA patients with high disease activity. Previous studies have shown that the TG plasma level is significantly elevated in RA patients [29], while reductions in total CH as well as CE in RA patients and other inflammatory conditions have been reported [30,31]. Furthermore, a high catabolic rate in RA patients leading to low CE plasma levels has been described, which, interestingly, could be reversed by tofacitinib treatment [32]. Despite the fact that we found similar trends with both of our platforms, the increase in total TG and reduction in total CE and PL plasma levels in RA patients were not significant in our case (Figure 4A). In turn, these alterations should be investigated using additional information about lipid species and lipoprotein subfraction. Combined lipid species–lipoprotein sub-particles’ analysis could improve our understanding of RA pathophysiology. 

In order to investigate if the diseased condition alters the correlation of a particular lipid–lipoprotein pair, we calculated the correlation difference between healthy controls and RA patients (Figure 4B) (interactive heatmap is available in Appendix A). The value zero indicates that there is no difference and positive and negative values indicate that the correlation is altered owing to the diseased condition. Moreover, the degree of difference indicates the degree of alteration. To make this more clear, we plotted the correlation of a few lipid–lipoprotein pairs in healthy controls and RA patients separately. For example, the correlation between L6TG and TG56:1-FA16:0 is negative in healthy controls, but positive in RA patients (Figure 4C). In contrast, the correlation between L4PL and PI 18:0_20:3 is positive in healthy controls and negative in RA patients. As a result, the analysis could be useful to discover changes in phenotypes due to lipid–lipoprotein alterations in RA patients and possibly discover novel correlation biomarkers. Therefore, combined lipid and lipoprotein measurements provide a broader map of lipid and lipoprotein metabolism and complement each other to obtain a better understanding of disease pathologies.

## 4. Conclusions

We have developed a combined workflow for subsequent NMR and DMS-SLA analysis on single plasma aliquots and demonstrate agreement between the platforms for lipid class quantification and differentiation of control and RA plasma materials. To demonstrate the value of combined NMR/DMS-SLA analysis, we carried out correlation analysis between lipoproteins and lipid species. We confirmed known correlations (e.g., VLDL–TG) and present a rich dataset for further data mining. To further substantiate the value of this analysis, we present potential novel biomarker candidates based on lipoprotein–lipid species correlation. Although biomarker discovery and validation will demand additional work, we show here that lipoprotein–lipid species correlation might serve as potential biomarkers and grant novel insights into pathophysiological mechanism. In summary, combined NMR/DMS-SLA analysis is feasible and provides added value when compared with the individual platforms.

## Figures and Tables

**Figure 1 metabolites-12-01030-f001:**
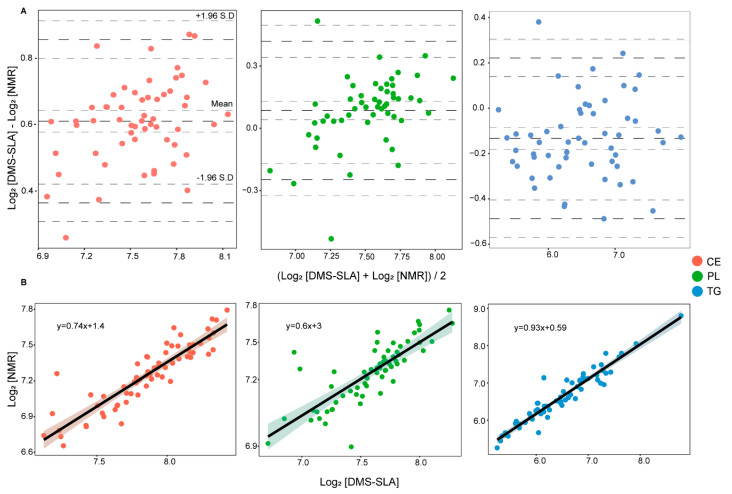
(**A**) Bland–Altman analysis of CE, PL, and TG concentrations from subsequent NMR and DMS-SLA measurements on single sample aliquots. X-axis and Y-axis represent mean and difference of the two platforms, respectively. The bold dotted lines represent the mean difference and limits of agreement, the dotted lines represent confidence intervals. (**B**) Pearson’s correlation analysis of CE, PL, and TG concentrations measured with B.I.LISA and DMS-SLA.

**Figure 2 metabolites-12-01030-f002:**
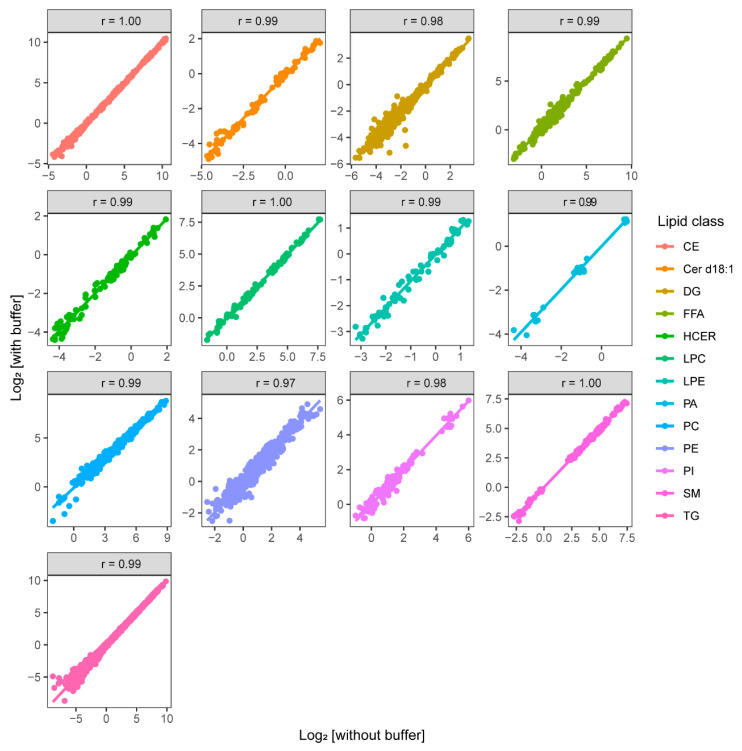
Pearson’s correlation analysis of lipid species concentrations across 13 detected lipid classes in plasma samples (with and without buffer addition) measured with DMS-SLA.

**Figure 3 metabolites-12-01030-f003:**
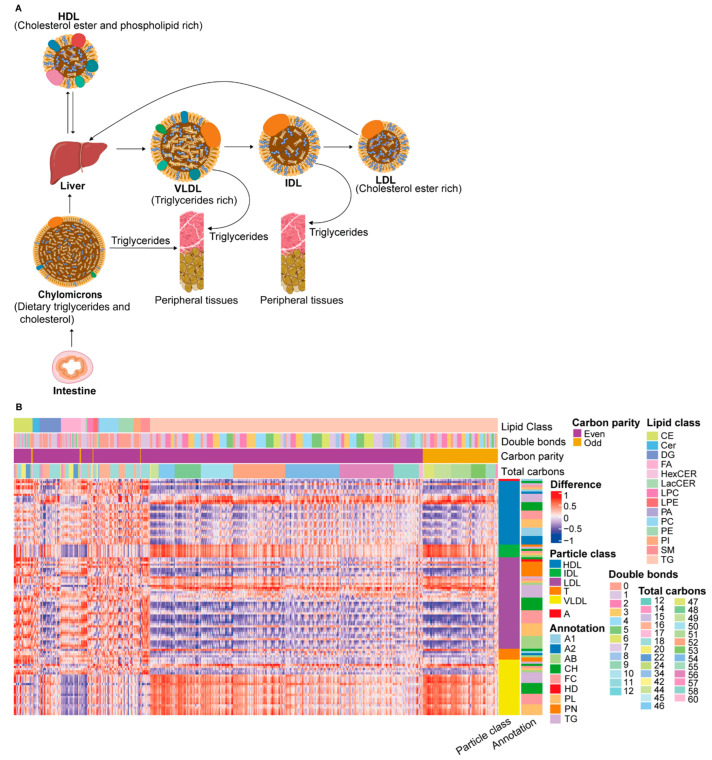
(**A**) Overview of lipoprotein particles and metabolism. Exogenous and endogenous lipoprotein pathway. (**B**) Correlation between lipids and lipoprotein particles in healthy controls. Particle annotations include lipid classes, particle number, and apo-proteins (PN = particle number; HD = LDL/HDL cholesterol ratio; FC = free cholesterol; CH = total cholesterol; AB = Apo-B; A2 = Apo-A2 A1 = Apo-A1; T = totals over all classes; A = ApoB100/Apo-A1 ratio) (full names of lipoproteins are available in Appendix A and the interactive heatmap in Appendix A).

**Figure 4 metabolites-12-01030-f004:**
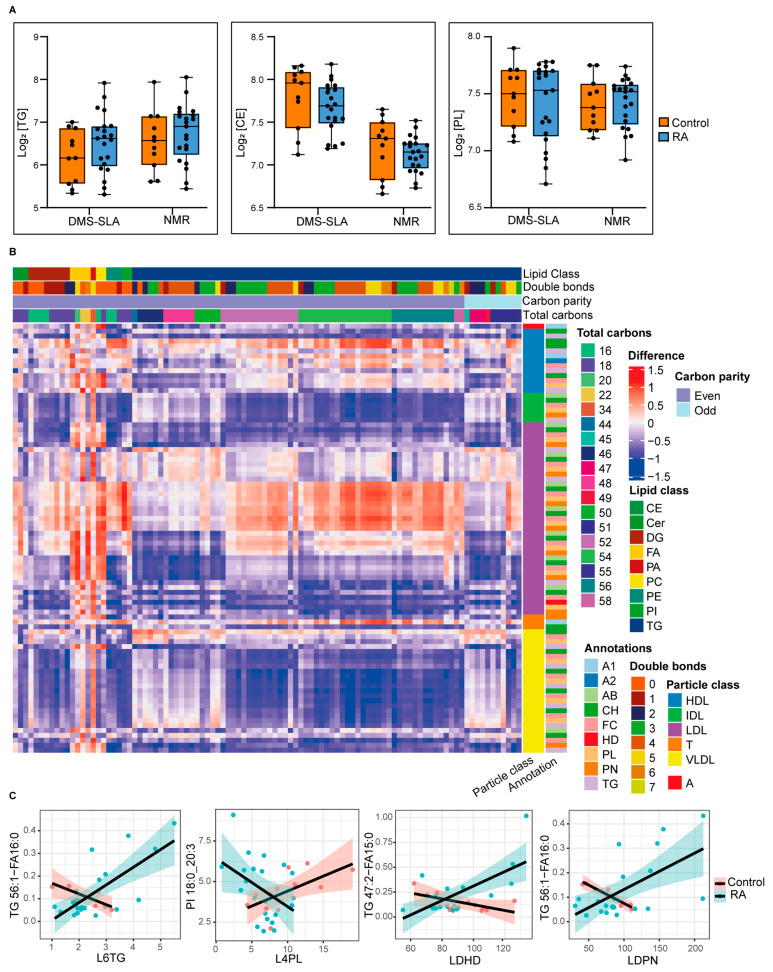
(**A**) Comparison of TG, CE, and PL concentrations in plasma samples from healthy controls and RA patients. The median value is marked by a horizontal line inside the box. The whiskers extending from top and bottom of the box represent the largest and smallest values, respectively. (**B**,**C**) The correlation difference of lipid–lipoprotein pairs among healthy controls and RA patients.

**Table 1 metabolites-12-01030-t001:** Pearson’s and Spearman’s correlation coefficients, *p*-values of correlation, and root-mean-square deviation (RMSD) with its normalized value (NRMSD). RMSD is calculated as the SD of residuals and NRMSD is calculated as the percentage ratio of RMSD to the mean measured value with NMR.

Lipid Class	Concentration	Pearson r	*p*-Value	Spearman ρ	*p*-Value	Mean (DMS-SLA)	Mean (NMR)	RMSD	NRMSD
**Triacyl glycerides**	**mg/dL**	0.97	1.4 × 10^−38^		5.04 × 10^−32^	102.1	111.39	16.5	14.82%
	**log_2_ [mg/dL]**	0.95	6.15 × 10^−34^	0.95	5.04 × 10^−32^	6.48	6.62	0.20	3.07%
**Cholesterol esters**	**mg/dL**	0.89	8.06 × 10^−23^	0.92	1.55 × 10^−26^	237.41	154.79	12.03	7.77%
	**log_2_ [mg/dL]**	0.89	8.06 × 10^−23^	0.92	1.55 × 10^−26^	7.86	7.25	0.12	1.62%
**Phospholipids**	**mg/dL**	0.81	1.56 × 10^−15^	0.81	1.03 × 10^−15^	197.46	183.75	17.36	9.44%
	**log_2_ [mg/dL]**	0.79	2.30 × 10^−14^	0.81	1.03 × 10^−15^	7.59	7.50	0.146	1.95%

## Data Availability

Data is available only on request due to privacy restrictions. The data are not publicly available due to the privacy of the participants of the LAIKO study.

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
