# Peer review of "Congruence and Complementarity of Differential Mobility Spectrometry and NMR Spectroscopy for Plasma Lipidomics"

_metabolites, 2022, doi:10.3390/metabo12111030_

Round 1

Reviewer 1 Report

This is a well written and concise side-by-side comparison of Bruker's NMR based lipoprotein subclass analysis with differential mobility spectrometry based shotgun lipidomics on the Sciex Lipidyzer platform, which underscores the respective merits of the two platforms and the potential merits of their combined application. The feasibility of subjecting the same plasma sample to successive NMR and MS analysis, the observed agreement of CE, TG and PL lipid measurements, and the complementary information to be gained from the combined application of NMR and MS are of interest to those working in the lipidomics field.

Author Response

We thank the reviewer for his/her positive comments.

Reviewer 2 Report

The authors have described  a combined workflow for subsequent NMR and DMS-SLA analysis on single plasma aliquots and demonstrate agreement between the platforms concerning class quantification and significant differentiation of control and RA plasma materials.

The present work is undoubtedly valuable and should be published, although I have some questions that need to be clarified.

Why IS was used only in DMS analysis instead of NMR. Is it some special purpose ?

Or explain why authors did not described using IS with NMR analysis sins (IS kit) include deuterated standards.

Which transitions were used for particular analysis in the MRM method. ? It might  be added to supplementary material or indicate the exact source of such transitions (database or publication)

Positive/negative  mode in MS/MS analysis detected wich classes of lipids?

Author Response

IS:

For FIA-DMS-MRM analysis a IS is necessary for quantification due to known pitfalls of ESI in bioanalysis, e.g. matrix effects. With respect to the use of internal standards in NMR analysis we would like to point out that NMR is an absolute quantitative technology based on the integration of the obtained 1H-integrals. In this respect, NMR does not necessitate the addition of an internal standard for quantification. Moreover, classic pitfalls of LC-MS quantification such as matrix effects do not play a significant role for NMR spectroscopy. Nevertheless, a standard was used, but it was an external standard: the QuantRefC standard sample that was measured before each series of measurement. The QuantRefC sample is supplied by Bruker and is a solution of a number of compounds of known concentrations, sealed in an NMR tube. The IVDr-BILISA addon to the Bruker TopSpin software then uses the spectrum of the QuantRefC sample for the calibration of the plasma spectra. This is a standard part of the BILISA measurement protocol that is provided by Bruker. This is mentioned concisely in section 2.4 of the manuscript.

Transitions and pos/neg ionization: we added a statement that these and additional detailed information about the method can be found in refs 4 and 7.